# Endoscopic Combined Intrarenal Surgery Versus Percutaneous Nephrolithotomy for Complex Renal Stones: A Systematic Review and Meta-Analysis

**DOI:** 10.3390/jpm12040532

**Published:** 2022-03-28

**Authors:** Yung-Hao Liu, Hong-Jie Jhou, Meng-Han Chou, Sheng-Tang Wu, Tai-Lung Cha, Dah-Shyong Yu, Guang-Huan Sun, Po-Huang Chen, En Meng

**Affiliations:** 1Department of Surgery, Division of Urology, Tri-Service General Hospital, National Defense Medical Center, Taipei 114, Taiwan; michael40315@yahoo.com.tw (Y.-H.L.); princecharmingben@gmail.com (M.-H.C.); doc20283@gmail.com (S.-T.W.); tailungcha@gmail.com (T.-L.C.); yuds45@gmail.com (D.-S.Y.); ghsun@gmail.com (G.-H.S.); 2Department of Neurology, Changhua Christian Hospital, Changhua 500, Taiwan; xsai4295@gmail.com; 3Department of Internal Medicine, Tri-Service General Hospital, National Defense Medical Center, Taipei 114, Taiwan

**Keywords:** endoscopic combined intrarenal surgery, percutaneous nephrolithotomy, complex renal stones, stone-free rates, safety, efficiency

## Abstract

Background: Endoscopic combined intrarenal surgery (ECIRS) adds ureteroscopic vision to percutaneous nephrolithotomy (PCNL), which can be helpful when dealing with complex renal stones. Yet, there is still no consensus on the superiority of ECIRS. We aimed to critically analyze the available evidence of studies comparing efficacy, safety, bleeding risk, and efficiency of ECIRS and PCNL. Methods: We searched for studies comparing efficacy (initial and final stone-free rate), safety (postoperative fever, overall and severe complications), efficiency (operative time and hospital stay) and bleeding risk between ECIRS and PCNL. Meta-analysis was performed. Results: Seven studies (919 patients) were identified. ECIRS provided a significantly higher initial stone-free rate, higher final stone-free rate, lower overall complications, lower severe complications, and lower rate of requiring blood transfusion. There was no difference between the two groups in terms of postoperative fever, hemoglobin drop, operative time, and hospital stay. In the subgroup analysis, both minimally invasive and conventional ECIRS were associated with a higher stone-free rate and lower complication outcomes. Conclusions: When treating complex renal stones, ECIRS has a better stone-free rate, fewer complications, and requires fewer blood transfusions compared with PCNL. Subgroups either with minimally invasive or conventional intervention showed a consistent trend.

## 1. Introduction

Renal stone is a common disorder, and complex renal stones are defined as having multiple stones or having anatomical or functional abnormalities, regardless of being peripheral or branched stones. Staghorn stones with their branching characteristics, occupying the renal pelvis and one or more calices, are the most complicated type. They usually have large stone burdens, determined by the number, diameter, and location of stones evaluated on images [1]. Since its development in 1976, percutaneous nephrolithotomy (PCNL) has been the indicated treatment for these cases with stone-free rates (SFRs) of 98.5% and 71% for partial and complete staghorn stones, respectively [2].

However, in cases with greater stone burden, PCNL is not the only option. In 1992, Dr. JG Ibarluzea utilized the clear visual field of the ureteroscope to remove stone fragments through an Amplatz sheath while performing PCNL simultaneously [3]. Later in 2008, Dr. CM Scoffone coined the term endoscopic combined intrarenal surgery (ECIRS) and operated under the Galdakao-modified supine Valdivia (GMSV) position, an adaption of the prone position [4].

ECIRS aimed to improve the one-step resolution of urolithiasis while reducing the number of access tracts [5]. Multiple retrospective studies comparing ECIRS and PCNL have reported contradictory outcomes. There is still no consensus on the superiority of ECIRS in terms of operative time, hospital stay, and even stone free rate or complications.

Furthermore, as techniques for miniaturized access in urolithiasis evolved, the mini-percutaneous access system (14–20 Fr sheath size) has been widely adopted. We have also conducted subgroup analysis for patients who underwent conventional-PCNL (cPCNL) or mini-PCNL (mPCNL) to compare the two procedures. This meta-analysis aims to compare the efficacy, safety, and efficiency between ECIRS and PCNL in patients with complex renal stones in order to provide recommendations for physicians in clinical practice.

## 2. Materials and Methods

### 2.1. Study Design

The study follows the Preferred Reporting Items for Systematic Review and Meta-analysis (PRISMA) and the Meta-analysis of Observational Studies in Epidemiology (MOOSE) [6,7] statement (Appendix A and Appendix B). The study is also registered in the Open Science Framework (OSF, DOI: 10.17605/OSF.IO/DRBFZ).

### 2.2. Search Strategy

From the inception through June 2021, databases including the Cochrane Library, PubMed, and Embase were searched. We conducted the search using subject headings and search field tags of the title, abstract, and keywords, comprised of “endoscopic combined intrarenal surgery” and “percutaneous nephrolithotomy” (details in Appendix C).

### 2.3. Eligibility Criteria

Studies that met all the following inclusion criteria were selected: (1)Types of participants: patients with complex renal stone.(2)Types of interventions: Studies comparing ECIRS and PCNL were eligible.(3)Types of outcome measures: Our outcomes of interest are categorized into “efficacy”, “safety”, and “efficiency”. Studies that reported at least an outcome of interest (i.e., initial stone free rate) were included.

Moreover, the studies should provide adequate information to calculate the effect estimated for meta-analysis. We did not exclude studies based on publication date, language, or geographical area. The exclusion criteria were as follows: overlapping or duplicate publication; studies in which necessary data could not be extracted; reviews, letters, and case reports.

### 2.4. Risk of Bias Assessment

The quality of the RCTs was appraised using the Cochrane Handbook for Systematic Reviews of Interventions, Risk of Bias Tool [8]. We also used the Newcastle–Ottawa Scale for the quality of prospective non-randomized studies [9] (Appendix D).

### 2.5. Data Extraction and Outcome Measurement

Two reviewers (Y.-H. Liu and P.-H. Chen) independently extracted datasets from the eligible studies. There were nine outcomes in the current study defined as follows:

#### 2.5.1. Efficacy Outcomes

(1)Initial stone-free rate (Initial SFR): Absence of stone or residual stone fragments on plain abdominal X-ray (Kidney–Ureter–Bladder, KUB) or non-contrasted abdominal computed tomography (NCCT) within 4 weeks post-operation, or as defined by each study.(2)Final stone-free rate (Final SFR): The stone-free status was defined as above, but was assessed after the auxiliary procedure (i.e., shock wave lithotripsy, PCNL or ureteroscopic lithotripsy).

#### 2.5.2. Safety Outcomes

(1)Overall complications: Perioperative complications were graded according to the Clavien classification system. Overall complications included all grades.(2)Severe complications: Clavien–Dindo classification system ≥grade 2.(3)Postoperative fever: Transient body temperature taken >38.5 °C after operation.

#### 2.5.3. Bleeding Risk

(1)Hemoglobin drop: The postoperative hemoglobin level decreased comparing with that of pre-operative evaluation.(2)Required blood transfusion: Blood transfusion needed due to significant hemorrhage.

#### 2.5.4. Efficiency outcomes

(1)Operative time: Time taken on the operating table, from positioning to the end of the procedure.(2)Hospital stay: Number of days since admission for pre-operative evaluation, operation, imaging for SFR assessment, and the treatment if complications occurred.

### 2.6. Quality Assessment

The Grading of Recommendations Assessment, Development, and Evaluation (GRADE) methodology was used to assess the certainty of evidence from the included studies (Appendix E) [10].

### 2.7. Subgroup Analysis, Meta-Regression, and Sensitivity Analysis

A priori subgroup analysis explored the influence of miniaturized access or conventional access of the operation on the pooled effect estimates (Appendix F). We performed a mixed-effects meta-regression analysis to evaluate the potential influence of publication date and Amplatz sheath size on the heterogeneity for the outcomes. We assessed the robustness of treatment effects on outcomes via a sensitivity analysis [11] that excluded high-risk-of-bias cohort studies (Appendix H).

### 2.8. Statistical Analysis

We analyzed dichotomous variables by calculating odds ratios (ORs) with 95% confidence intervals (CIs). The continuous variables were estimated with the mean difference (MD). Both dichotomous and continuous outcomes were calculated using the inverse variance method. We reported both random-effects meta-analysis models with the DerSimonian–Laird estimator and fixed-effect model. Statistical heterogeneity was assessed using Cochran’s Q statistic and quantified by the I^2^ statistic [12].

Publication bias was evaluated using funnel plots and Egger’s test. (Appendix I) [13] All statistical analyses were performed using the “metaphor” and “meta” [14,15] packages of R software version 4.1.0.

To obtain conclusive results [16], trial sequential analysis (TSA) was applied to calculate the diversity-adjusted required information size (RIS) and trial sequential monitoring boundaries (Appendix J). The models for all outcomes were based on an alpha of 5% and a power of 80%. TSA was performed using TSA software version 0.9.5.10 Beta (Copenhagen Trial Unit, Copenhagen, Denmark).

## 3. Results

### 3.1. Study Identification and Selection

The search flow diagram is shown in Figure 1. Seven studies were included in the meta-analysis.

### 3.2. Study Characteristics and Risk of Bias Assessment

Table 1 illustrates the characteristics of the seven included studies [17,18,19,20,21,22,23]. The risk of bias assessment is shown in Appendix D.

### 3.3. Outcomes

#### 3.3.1. Efficacy Outcome

##### Initial Stone Free Rate (Initial SFR)

The outcome of initial SFR was reported in all seven studies [17,18,19,20,21,22,23], which included 401 and 521 patients in the ECIRS and PCNL groups, respectively (Figure 2). The initial SFR was significantly higher in ECIRS patients than in PCNL patients (random-effects, OR 3.50; 95% CI 2.16–5.67; I^2^  =  47%, Cochran’s Q test *p*-value = 0.08). In TSA, the cumulative number of patients exceeded the required information size of 243 and the Z-curves surpassed the significance boundary in favor of ECIRS, suggesting conclusive results and providing convincing statistical evidence to our findings. The TSA-adjusted confidence interval was OR 3.50 with 95% CI 1.87–6.53 (Appendix J).

##### Final Stone Free Rate (Final SFR)

The outcome of final SFR was reported in five studies [17,19,20,22,23], which included 271 and 350 patients in the ECIRS and PCNL groups, respectively (Figure 2). The final SFR was significantly higher in ECIRS patients than in PCNL patients (random-effects, OR 3.06; 95% CI 1.57–5.59; I^2^  =  37%, Cochran’s Q test *p*-value = 0.17). In TSA, the cumulative number of patients exceeded the required information size of 304 and the Z-curves surpassed the significance boundary in favor of ECIRS, suggesting conclusive results and providing convincing statistical evidence to our findings. The TSA-adjusted confidence interval was OR 3.06 with 95% CI 1.19–7.85 (Appendix J).

#### 3.3.2. Safety Outcome

##### Overall Complications

The overall complication outcome was reported in seven studies [17,18,19,20,21,22,23], which included 401 and 521 patients in the ECIRS and PCNL groups, respectively (Figure 3). Patients with overall complications were significantly fewer in the ECIRS group than in the PCNL group (random-effects, OR 0.45; 95% CI 0.29–0.70; I^2^  =  31%, Cochran’s Q test *p*-value = 0.19). In TSA, the cumulative number of patients exceeded the required information size of 675 and the Z-curves surpassed the significance boundary in favor of ECIRS, suggesting conclusive results and providing convincing statistical evidence to our findings. The TSA-adjusted confidence interval was OR 0.45 with 95% CI 0.28–0.72 (Appendix J).

##### Severe Complications

The outcome of severe complications was reported in six studies [17,18,19,21,22,23], which included 328 and 423 patients in the ECIRS and PCNL groups, respectively (Figure 3). The number of patients with severe complications was significantly fewer in the ECIRS group than in the PCNL group (random-effects, OR 0.29; 95% CI 0.16–0.52; I^2^  = 0%, Cochran’s Q test *p*-value = 0.94). In TSA, the cumulative number of patients exceeded the required information size of 500, and the Z-curves surpassed the significance boundary in favor of ECIRS, suggesting conclusive results and providing convincing statistical evidence to our findings. The TSA-adjusted confidence interval was OR 0.29 with 95% CI 0.15–0.56 (Appendix J).

##### Postoperative Fever

The outcome of postoperative fever was reported in five studies [17,18,19,22,23], which included 265 and 327 patients in the ECIRS and PCNL groups, respectively (Figure 3). The incidence of postoperative fever was not significantly different between ECIRS and PCNL patients (random-effects, OR 0.65; 95% CI 0.34–1.24; I^2^ = 17%, Cochran’s Q test *p*-value = 0.31). In TSA, the cumulative number of patients did not exceed the required information size of 2822, and the Z-curves did not surpass any significance boundary either, suggesting inconclusive results. Further studies are needed to provide convincing statistical evidence. The TSA-adjusted confidence interval was OR 0.65 with 95% CI 0.14–2.97 (Appendix J).

#### 3.3.3. Bleeding Risk

##### Hemoglobin Drop

The outcome of hemoglobin drop was reported in five studies [18,19,21,22,23], which included 295 and 389 patients in the ECIRS and PCNL groups, respectively (Figure 4). The incidence of hemoglobin drop was not significantly different between ECIRS and PCNL patients (random-effects, MD −0.80 g/dL; 95% CI −1.64–0.04; I^2^ = 98%, Cochran’s Q test *p* value < 0.01). In TSA, the cumulative number of patients did not exceed the required information size of 1658, and the Z-curves did not surpass any significance boundary either, suggesting inconclusive results. Further studies are needed to provide convincing statistical evidence. The TSA-adjusted confidence interval was MD −0.80 with 95% CI −2.22–0.63 g/dL (Appendix J).

##### Required Blood Transfusion

The outcome of required blood transfusion was reported in six studies [17,18,19,21,22,23], which included 328 and 423 patients in the ECIRS and PCNL groups, respectively (Figure 4). The number of required blood transfusions was lower among ECIRS patients than in PCNL patients (random-effects, OR 0.33; 95% CI 0.12–0.91; I^2^ = 0%, Cochran’s Q test *p*-value 0.98). In TSA, the cumulative number of patients did not exceed the required information size of 799, and the Z-curves only surpassed the traditional significance boundary in favor of ECIRS but not the TSA monitoring boundary, suggesting inconclusive results. Further studies are needed to provide convincing statistical evidence. The TSA-adjusted confidence interval was OR 0.33 with 95% CI 0.10–1.02 (Appendix J).

#### 3.3.4. Efficiency Outcome

##### Operative Time

The outcome of operative time was reported in six studies [17,18,19,21,22,23], which included 328 and 423 patients in the ECIRS and PCNL groups, respectively (Figure 5). Operative time was not significantly different between ECIRS and PCNL patients (random-effects, MD −6.73 min; 95% CI −19.91–6.46; I^2^  = 91%, Cochran’s Q test *p*-value < 0.01). In TSA, the cumulative number of patients did not exceed the required information size of 5901, and the Z-curves did not surpass any significance boundary either, suggesting inconclusive results. Further studies are needed to provide convincing statistical evidence. The TSA-adjusted confidence interval was MD −6.73 with 95% CI −60.55–47.10 min (Appendix J).

##### Hospital Stay

The outcome of hospital stay was reported in six studies [17,18,19,20,22,23], which included 338 and 425 patients in the ECIRS and PCNL groups, respectively (Figure 5). The length of hospital stay was not significantly different between ECIRS and PCNL patients (random-effects, MD −2.05 days; 95% CI −4.14–0.05; I^2^  = 94%, Cochran’s Q test *p*-value < 0.01). In TSA, the cumulative number of patients did not exceed the required information size of 1646 and the Z-curves did not surpass any significance boundary either, suggesting inconclusive results. Further studies are needed to provide convincing statistical evidence. The TSA-adjusted confidence interval was MD −2.05 with 95% CI −5.37–1.28 days (Appendix J).

### 3.4. Subgroup Analysis in Different Procedure Types and Study Types

In subgroup analysis, patients receiving mECIRS versus mPCNL (Appendix F) had higher initial SFR, higher final SFR, fewer overall complications, fewer severe complications, shorter hospital stay and lower incidence of postoperative fever, but no difference in operative time, incidence of hemoglobin drop and patients requiring blood transfusion. Besides, in the subgroup analysis of different study types (Appendix G), the results from retrospective cohort studies did not alter the trend of meta-analysis results in all the outcomes.

### 3.5. Meta-Regression

In the meta-regression, there was no difference in the interaction of the publication date and radius access length with all the outcomes, which indicated that the heterogeneities of the publication date (Appendix K) and Amplatz sheath size (Appendix L) did not influence the results of meta-analysis.

### 3.6. Sensitivity Analysis

In the sensitivity analysis, the pooled estimates within the 95% CI were maintained after excluding the highest risk-of-bias cohort studies (i.e., Newcastle–Ottawa quality assessment score ≤ 7) across all the results for these outcomes (Appendix H).

Furthermore, we performed a stepwise sensitivity analysis to exclude the high risk-of-bias cohort studies (i.e., Newcastle–Ottawa quality assessment score ≤ 8). The pooled estimates within the 95% CI were maintained across all the results for these outcomes, except for required blood transfusion (Appendix I).

## 4. Discussion

To our knowledge, this is the first systematic review and meta-analysis to investigate and compare the efficacy, safety, and efficiency of ECIRS and PCNL on patients with complex renal stones. We found that ECIRS improved both initial and final SFR, while lowering both overall and severe complications as well as the need for blood transfusion. No significant differences were found for the other complications (i.e., postoperative fever and hemorrhage) as well as for operative time and hospital stay.

Performing additional RIRS at the same time of PCNL contribute to the improvement of SFR by serving diagnostic and therapeutic functions, including supervision of renal access and the urinary tract below the kidney, avoidance of multiple percutaneous access tracks by endoscopic exploration of calices that were unreachable by nephroscopy, irrigation during lithotripsy, and passing the stone fragments through the Amplatz sheath [3].

To depict the cooperative relationship between RIRS and PCNL more vividly, the following example can be made. Two cases (2.7%) of patients undergoing PCNL monotherapy presented with steinstrasse, multiple stone fragments accumulating along the ureter after the surgery, in the study by F Zhao et al., while none was found in the ECIRS group [18]. In ECIRS, the fragments in the ureter could be pushed upward and extracted through the Amplatz sheath when coordinating the two pieces of equipment. Performed together, RIRS and PCNL have a synergistic effect and overcome their individual limitations.

The most concerning complications of PCNL are hemorrhage, infection, and thoracic complications (i.e., pneumothorax, hemothorax, etc.) [24]. RIRS concerns people the most with ureteral stent discomfort, ureteral wall injury, and stone migration [25]. Some may argue that ECIRS can possibly add up the risks of both PCNL and RIRS [4]; in fact, in our meta-analysis, ECIRS had significantly fewer overall and severe complications than PCNL. ECIRS patients also required less blood transfusions. As more excessive bleeding conditions necessitate more blood transfusions [26], our meta-analysis suggested that ECIRS causes fewer massive bleeding events. Compared with PCNL, the mean difference of hemoglobin drop in ECIRS group is −0.8 (−1.64; 0.04), which suggests there may be less blood loss. Although there are no significant differences, Zhao et al., Hamamoto et al., Leng et al., and Xu et al. all support such trend.

Subgroup analyses were performed to compare between the conventional group and the minimally invasive group. In 1976, Fernström and Johansson first invented cPCNL, also termed standard PCNL, which has a tract size ≥22 Fr [24]. On the other hand, in 1998, Jackman introduced the miniaturization of the instrument set (now termed mPCNL) for the treatment of nephrolithiasis in children then [27]. In our subgroup analysis (mECIRS vs mPCNL; cECIRS vs cPCNL), the SFR and complication rate were consistent with the primary outcome (ECIRS vs PCNL). In the minimally invasive subgroup (mECIRS vs mPCNL), mECIRS had significantly shorter hospital stays, according to de la Rosette, which was associated with lower Clavien-Dindo scores, implying fewer severe complications [28]. Moreover, mECIRS requiring fewer auxiliary procedures may also shorten hospital stay [18].

However, ECIRS is still not prevalent in clinical practice due to several concerns. First, requiring two endovision systems and cooperation between two surgeons can be an issue in limited-resource settings. Second, the problem of cost was mentioned in the study by Jung HD et al., wherein cases of unilateral renal stones are not allowed to require the cost of PCNL and RIRS at the same time, which may be a burden to the hospital in Korea [29]. In Taiwan, ECIRS costs an additional surgical fee and self-paid medical devices that are not covered by the national health insurance; this may influence the patients’ willingness to undergo the surgery. Third, with the combination of two procedures, operative time is sometimes considered longer in ECIRS. In fact, our meta-analysis revealed no significant difference in operative time between ECIRS and PCNL [17,20]. The concerns mentioned above should be reevaluated as we came to realize: the elimination of potential complications decreases the expense of rescue measures, such as blood transfusion or intravenous antibiotics, and the superior SFR eliminates the need for auxiliary procedures and their associated costs [30]. The advantages of ECIRS may outweigh its disadvantages to some extent.

In our meta-analysis, we have not only carefully screened and included the studies that met the aforementioned criteria, but also performed subgroup analyses to clarify the differences among minimally invasive and conventional groups separately. However, our study has some limitations. First, the number of patients included (*n* = 919) was relatively small, which may result from the fact that ECIRS is still not clinically prevalent. Second, six out of the seven included studies were not RCTs, which can possibly cause intrinsic bias. There was also no consensus on the patient’s position (prone, supine, or GMSV position) [3]. Heterogeneity may exist among the included studies. There was a paucity of RCTs that could have elucidated the comparative outcomes of ECIRS and other methods to remove complex renal stones. More detailed secondary outcomes can be obtained in future studies to cover our limitations.

## 5. Conclusions

In conclusion, our meta-analysis of the current evidence suggests that ECIRS is more effective and safer than PCNL. When treating complex renal stones, ECIRS has better initial/final SFR, fewer overall/severe complications, and requires fewer blood transfusions than PCNL. Both minimally invasive and conventional subgroups supported ECIRS in the SFR and complication outcomes. In the minimally invasive subgroup, ECIRS was favored due to shorter hospital stays and less postoperative fever. No significant differences were found in other outcomes, which require more high-quality studies to determine.

## Figures and Tables

**Figure 1 jpm-12-00532-f001:**
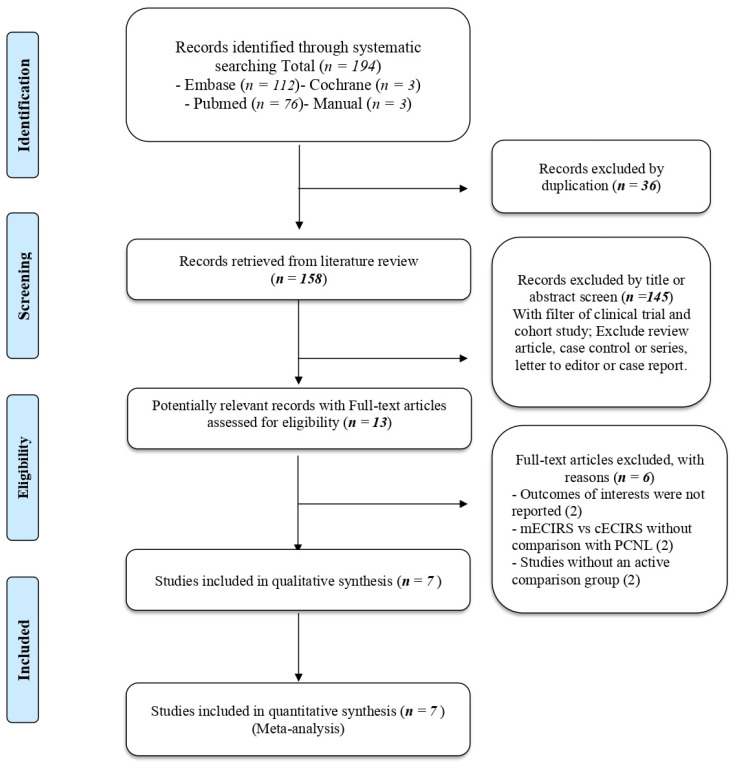
PRISMA flow diagram. mECIRS: mini-endoscopic combined intrarenal surgery, cECIRS: conventional endoscopic combined intrarenal surgery, PCNL: percutaneous nephrolithotomy.

**Figure 2 jpm-12-00532-f002:**
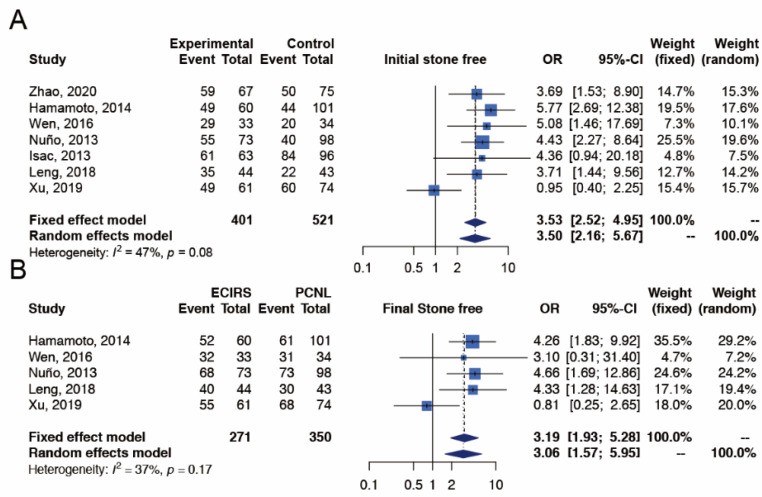
Meta-analysis of efficacy outcomes, including (**A**) initial stone free rate [17,18,19,20,21,22,23] and (**B**) final stone free rate [17,19,20,22,23] between ECIRS and PCNL groups.

**Figure 3 jpm-12-00532-f003:**
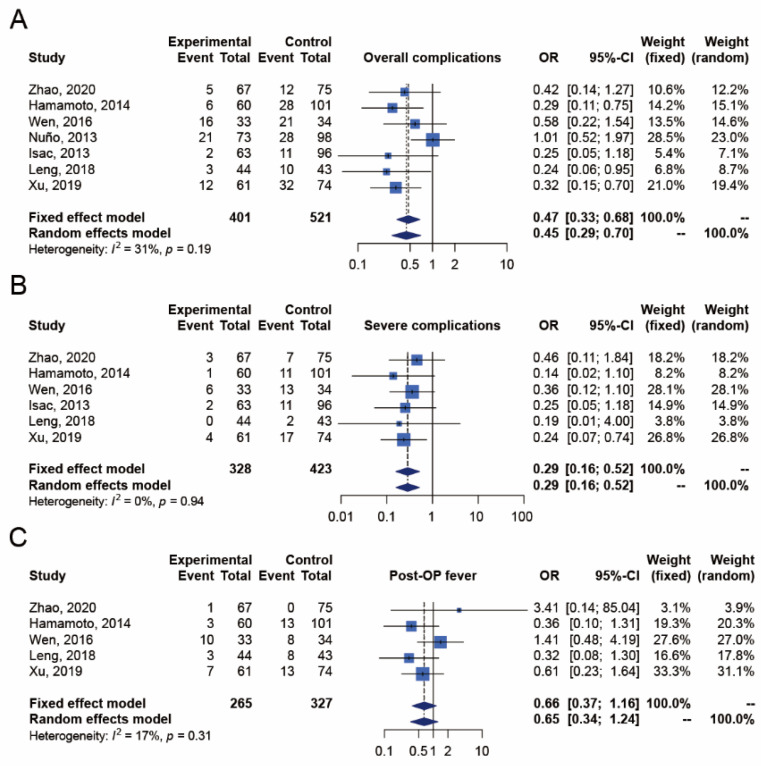
Meta-analysis of safety outcomes, including (**A**) overall complications [17,18,19,20,21,22,23], (**B**) severe complications [17,18,19,21,22,23], and (**C**) postoperative fever [17,18,19,22,23] between ECIRS and PCNL groups.

**Figure 4 jpm-12-00532-f004:**
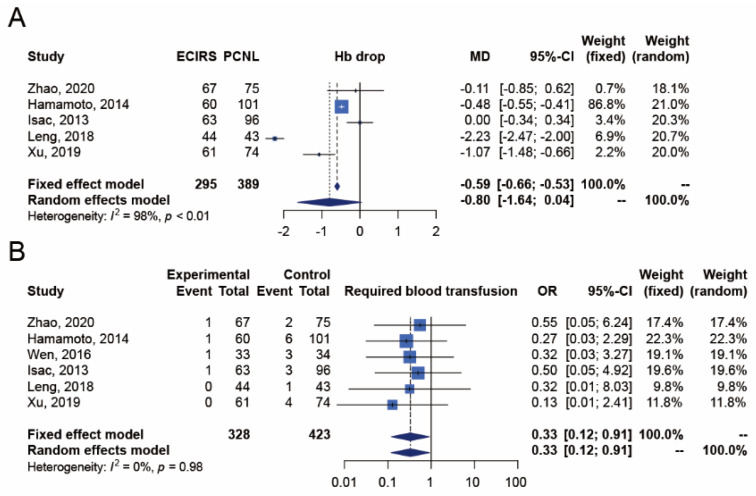
Meta-analysis of bleeding risks, including (**A**) hemoglobin drop [18,19,21,23] and (**B**) required blood transfusion [17,19,21,23] between ECIRS and PCNL groups.

**Figure 5 jpm-12-00532-f005:**
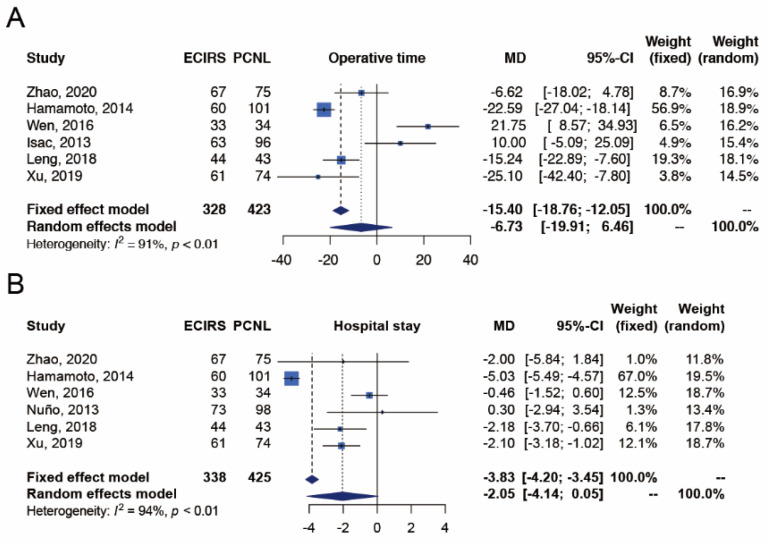
Meta-analysis of efficiency outcomes, including (**A**) operative time [17,18,19,21,22,23] and (**B**) hospital stay [17,18,19,20,22,23] between ECIRS and PCNL groups.

**Table 1 jpm-12-00532-t001:** Characteristics of included studies.

Author, Year	Country	Study Period	Study Design	No. of Patients	Age (Mean)	Male (%)	BMI (kg/m^2^)	Stone Burden Characteristics	No. of Staghorn Stone (%)	No. of Complete Staghorn stone (%)	Intervention	Comparator	Percutaneous Access Size	ECIRS Position	PCNL Position
Zhao, 2020 [17]	China	Jan 2018– Oct 2019	RCS	140	53.13	64.2	25.61	Area 700 mm^2^	16.4	8.4	mECIRS	mPCNL	16–18F	GMSV	prone
Hamamoto, 2014 [18]	Japan	Feb 2004– Jan 2013	RCS	161	53.17	75.8	24.62	Max 36.7 mm	35.4	17.4	mECIRS	mPCNL, cPCNL	(mini) 18F, (con) 30F	prone split-leg	prone
Wen, 2016 [19]	China	May 2012– Oct 2014	RCT	67	44.49	58.2	21.9	Area 667 mm^2^	100	NS	mECIRS	mPCNL	20F	GMSV	prone
Nuño, 2013 [20]	Spain	Jan 2005– Dec 2011	RCS	171	51.4	42.1	NS	Area 694.1 mm^2^	43.2	24.6	cECIRS	cPCNL	24–30F	GMSV	supine
Isac, 2013 [21]	USA	Aug 2010– Jan 2012	RCS	158	57.6	45.5	30.78	Cumulative 30.6 mm	NS	NS	cECIRS	cPCNL	30F	prone split-leg	prone
Leng, 2018 [22]	Japan	Feb 2004– Jan 2013	RCS	87	45.98	59.8	NS	Mean 52.2 mm	100	33.3	mECIRS	mPCNL	16–18F	oblique supine lithotomy	oblique supine lithotomy
Xu, 2019 [23]	China	NS	RCS	135	50.03	48.2	23.05	Mean 58.14 mm	100	65.19	mECIRS	mPCNL	16–22F	NS	NS

RCS: retrospective cohort studies; RCT: randomized control trial; mECIRS: minimally-invasive endoscopic combined intrarenal surgery; cECIRS: conventional endoscopic combined intrarenal surgery; mPCNL: minimally-invasive percutaneous nephrolithotomy; cPCNL: conventional percutaneous nephrolithotomy; F: French; GMSV: Galdakao-modified supine Valdivia; NS: not specified.

## Data Availability

Details of the checklists, literature search strategy, risk of bias assessment, subgroup analyses, sensitivity analyses, and trial sequential analysis are reported in the appendix.

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
