# Peer review of "Endoscopic Combined Intrarenal Surgery Versus Percutaneous Nephrolithotomy for Complex Renal Stones: A Systematic Review and Meta-Analysis"

_jpm, 2022, doi:10.3390/jpm12040532_

Round 1

Reviewer 1 Report

Line 74: the authors mentioned that they did not exclude the studies based on their publication date. However, according to lines 43 and 45 and the timing of procedures introduction for the first time, the publication date would be inevitably affected.

Line 96: the extension of postoperative fever recording should be mentioned.

Line 98: the hemoglobin drop needs to be discussed quantitatively.

Line 112: an should be omitted.

Line 155: The references should be separated.

Reviewer 2 Report

This review is original and very accurate. In general a well written and structured  review.  No corrections are required.

Reviewer 3 Report

Authors propose a meta-analysis comparing ECIRS to PCNL. They found some serious advantages for ECIRS. The manuscript is overall well written and methods are fine. I have some concercerns:

  1. authors mixed results from retrospective sutdies with those from a RCT. They should perform a subgroup analysis excluding the RCT.
  2. PCNL radius access was different in different studies, a meta-regression would be appropriate to test the effect of nephroscope on outcomes examined
  3. Please use PICOs in methods
